# Effect of Electromagnetic Pulses on the Microstructure and Abrasive Gas Wear Resistance of $Al_{0.25}CoCrFeNiV$ High Entropy Alloy

Olga Samoilova [1], Nataliya Shaburova [1], Valeriy Krymsky [2], Vyacheslav Myasoedov [3], Ahmad Ostovari Moghaddam [1,*] and Evgeny Trofimov [1]

[1] Department of Materials Science, Physical and Chemical Properties of Materials, South Ural State University, 76 Lenin Avenue, 454080 Chelyabinsk, Russia; samoilovaov@susu.ru (O.S.); shaburovana@susu.ru (N.S.); trofimovea@susu.ru (E.T.)
[2] Department of Theoretical Foundations of Electrical Engineering, South Ural State University, 76 Lenin Avenue, 454080 Chelyabinsk, Russia; krymskiivv@susu.ru
[3] Laboratory of Composite Materials, South Ural State University, 76 Lenin Avenue, 454080 Chelyabinsk, Russia; vmyasoedov74@mail.ru
[*] Correspondence: ostovarim@susu.ru

**Abstract:** High entropy alloys (HEAs) are among the most promising materials, owing to their vast chemical composition window and unique properties. Segregation is a well-known phenomenon during the solidification of HEAs, which negatively affects their properties. The electromagnetic pulse (EMP) is a new technique for the processing of a metal melt that can hinder segregation during solidification. In this study, the effect of an EMP on the microstructure and surface properties of $Al_{0.25}CoCrFeNiV$ HEA is studied. An EMP, with an amplitude of 10 kV, a leading edge of 0.1 ns, a pulse duration of 1 ns, a frequency of 1 kHz, and pulse power of 4.5 MW, was employed for melt treatment. It was found that the microstructure of $Al_{0.25}CoCrFeNiV$ HEA changes significantly from dendritic, for an untreated sample, to lamellar "pearlite-like", for an EMP treated sample. Moreover, EMPs triggered the formation of a needle-like σ-phase within the solid solution grains. Finally, these microstructural and compositional changes significantly increased the microhardness of $Al_{0.25}CoCrFeNiV$ HEA, from $343 \pm 10$ $HV_{0.3}$ (without the EMP) to $553 \pm 15$ $HV_{0.3}$ (after the EMP), and improved its resistance against gas-abrasive wear. Finally, an EMP is introduced as an effective route to modify the microstructure and phase formation of cast HEAs, which, in turn, opens up broad horizons for fabricating cast samples with tailorable microstructures and improved properties.

**Keywords:** electromagnetic pulses; high entropy alloys; segregation; microstructure; microhardness; gas-abrasive wear





## 1. Introduction

The concept of multicomponent alloys that typically consist of five or more elements with an equimolar (or near equimolar) ratio to ensure the highest entropy of mixing, was proposed by Cantor and Yeh in 2004 [1,2]. Further studies showed that a number of high entropy alloys (HEAs) based on transition metals have excellent mechanical characteristics, such as a combination of high strength and ductility [3,4], increased fatigue resistance [5,6], high hardness, and wear resistance [7–9]. The latter is of particular interest, since the replacement of steels and cast irons with HEAs can significantly extend the service life of the products working under high friction and wear conditions.

At present, the quest for optimal HEA compositions with increased wear resistance is still ongoing [9]. There are also alternative ways to improve the wear resistance of metals. Notably, studies are being carried out on the effect of reinforcement carbide particles on the tribological properties of HEAs [10,11]. It is also known that any changes in the microstructure of the resulting HEAs affect their friction and wear behaviors [12].

The effect of strong magnetic fields (MF) and the treatment of metal melts with electromagnetic pulses (EMP) are the latest techniques used to modify the microstructure of metals and alloys [13]. By using similar techniques for steels, cast irons, aluminum and copper alloys, finer microstructures [14–21]—and an almost two-fold increase in the microhardness of the treated samples [14,19–21]—have been realized. Moreover, Jie et al. [16] reported that MF treatment of commercially pure aluminum results in grain refinement and the transformation from columnar to equiaxed grains. It is also reported that nanosecond EMP treatment of aluminum cast alloys not only leads to an improvement in the microstructure, but also to a significant decrease in the porosity [17]. Ma et al. [18] reported that after a Cu–37.4% Pb melt was treated with electromagnetic pulses, not only did the size of the copper dendrites decrease after crystallization, but a more uniform distribution of lead inclusions also occurred, as well as an increase in its solubility in copper-rich solid solutions. In addition to microstructural changes and an increase in microhardness, an increase in strength and ductility has also been noted in the metal treated with EMP [22,23].

The effect of MF and EMP on the microstructure and properties of HEAs is discussed in a limited number of studies [24–27]. Guo et al. [24] showed that EMP treatment can change spinodal decomposition, improve ductility, and increase the corrosion pitting resistance of AlCoCrFeNi HEA. Zhao et al. [25] found out that after exposure to strong magnetic fields, the microstructure of AlCoCrFeNi HEA became coarse and more homogeneous. In addition, they found out that the volume percentage of the bcc phase increased, which improved the magnetic characteristics of the alloy. Deng et al. [26] drew attention to the fact that MF treatment of the AlCoCrCuFeNi alloy can suppress precipitation of Cu-rich fcc phase, and in turn, it can lead to an increase in the mechanical characteristics of the alloy. In addition, Guo et al. [27] reported a significant effect of EMP on the macro- and microstructure of CoCrFeNiCu HEA, which changes from a dendritic microstructure in the absence of pulses, to an almost equiaxed microstructure with an increase in the pulse frequency to 8 Hz. It was also found that EMP suppresses (to the point of complete absence) the segregation of copper into a separate phase in the treated samples. Moreover, EMP treatment increased the mechanical characteristics of the CoCrFeNiCu alloy; thus the microhardness index increased by a value of about 20% at a pulse frequency of 2 Hz.

It is known that vanadium can increase the microhardness and wear resistance of HEAs (in particular, $Al_x$CoCrFeNi HEAs); however, the microstructure of such samples is not ideal, since vanadium is prone to segregation, in some cases even in the form of coarse dendrites along the grain boundaries [28–30]. Based on the previously reported results on EMP treatment of non-ferrous alloys [20,22,23,31,32], dispersion of vanadium precipitates, and homogenization of the structure as a whole, can be expected after EMP treatment; therefore, the purpose of this work is to study the effect of high-power EMP treatment on the microstructure, microhardness, and gas-abrasive wear behavior of $Al_{0.25}$CoCrFeNiV HEA.

## 2. Materials and Methods

Samples of $Al_{0.25}$CoCrFeNiV HEA were fabricated by induction melting in a reducing atmosphere using high purity metals (granules and powders, >99.9 wt.%) according to the procedure described here [30].

The EMP treatment technique of metal melts is described in detail elsewhere [31,32], and we briefly explain it here. To generate electromagnetic pulses, we used a generator manufactured by FID Technology (St. Petersburg, Russia), which is presented in Figure 1a with the following characteristics: pulse amplitude of 10 kV, wherein the leading edge of the pulse was 0.1 ns, the pulse duration was 1 ns, the pulse repetition rate was 1 kHz, and the calculated pulse power was 4.5 MW. A distinctive feature of the generator used is its low power consumption of 50 W. The loss factor K was measured by using a picosecond pulsed reflectometer P5-15 (LLC Eliz, Cherepovets, Russia). The operating frequency band of the instrument is 0–5 GHz. The essence of the measurement is to determine the reflection from the end of the line that includes the cable (supplies energy from the generator), which wires to the electrode, and the electrode wires to the crucible. The measured value in the

cold state (without immersing the electrodes into the melt) was 0.3. This means that 0.3 of the power supplied from the generator is lost in the line elements (cable, wires, electrode). Consequently, the remaining 70% of the power gets into the melt.

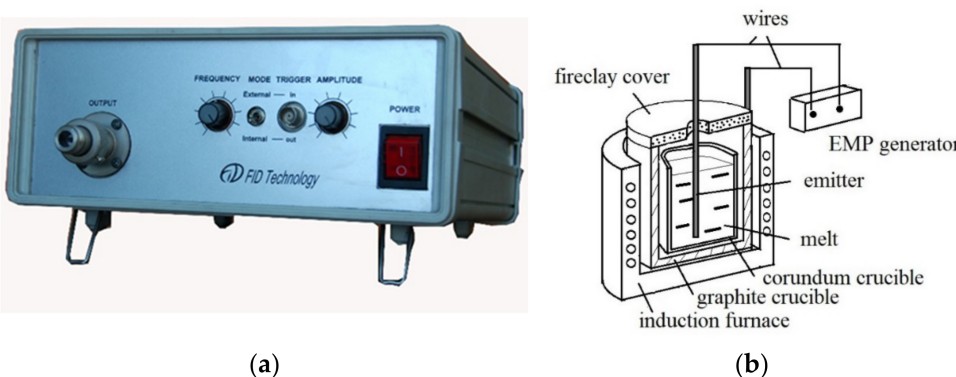

**Figure 1.** (**a**) EMP generator; (**b**) a schematic of the nanosecond EMP treatment setup.

Two independent experiments were carried out, in which a stoichiometric amount of metal powders were mixed and loaded into a corundum crucible and melted inside an induction furnace under a reducing atmosphere. The melt temperature was controlled by a portable pyrometer. During melting with pulse processing, when the melt was heated to a temperature of 1700 °C, one tungsten electrode was dipped into the melt and the second electrode was placed on the body of the graphite crucible (Figure 1b). EMP treatment of the melt was carried out for 5 min. For melting without impulse action, when the melt reached the same temperature, a W electrode was immersed into the melt and the melt was held for 5 min without turning on the generator. During the second melting (without exposure to electromagnetic radiation), the electrode was immersed into the melt to simulate the influence of introducing a cold electrode into the melt crystallization. In both cases, after five minutes of exposure, the corundum crucible with the melt was removed from the furnace and cooled by air.

The microstructure was studied using an Axio Observer D1.m optical inverted metallographic microscope equipped with a Thixomet Pro image analysis software and hardware system. Marble's reagent was used to reveal the microstructure. The microstructure of the longitudinal section of the ingots, and the surface morphology after gas-abrasive wear, were studied on a scanning electron microscope (SEM) JEOL JSM-7001F (JEOL, Tokyo, Japan), equipped with an energy-dispersive X-ray spectroscopy detector (EDS; Oxford INCA, Abingdon, Great Britain). An X-ray diffraction phase analysis (XRD) was carried out on a Rigaku (Tokyo, Japan) Ultima IV diffractometer using Cu–K$\alpha$ radiation ($\lambda$ = 0.15406 Å).

The microhardness was measured using an FM-800 microhardness tester (Future-Tech Corp., Tokyo, Japan) at a load of 300 g. To average the microhardness index, at least 18 measurements were used on each sample.

Gas abrasive wear tests were carried out using a Contracor CAB 135S abrasive blast cabin manufactured by Contracor GmbH (Wuppertal, Germany). Electrocorundum $Al_2O_3$ particles with a size of 100 μm were used as the abrasive material. For the gas-abrasive wear test, bars of 18 mm ×10 mm ×10 mm in size were fabricated from the HEA ingots, and an area of about 180 mm$^2$ was subjected to a wear test. An air jet with abrasive material was supplied into the cabin using an abrasive blast gun with a nozzle diameter of 8 mm under a pressure of 0.5 MPa. The distance from the nozzle to the surface of the sample was 150 mm. The abrasion resistance was evaluated using the mass loss parameter. To do this, the cast bar was first weighed, then gas-abrasive treatment was carried out over an interval of two minutes to weight the samples and record the weight loss. The total exposure time to the abrasive particles for each sample was about 20 min (10 weight loss measurements were recorded). Before weighting, each sample was washed in alcohol and blown with a compressor to clean the surface of abrasive particles that may affect the

weight loss of the samples. An electronic laboratory balance VK-1500, with an accuracy of 0.02 g, manufactured by CJSC Massa-K (St. Petersburg, Russia), was used for weighting.

## 3. Results and Discussion

### 3.1. Microstructure

Figure 2 shows the optical microscope images of the microstructure of the etched samples. As shown in Figure 2, the microstructure of $Al_{0.25}CoCrFeNiV$ alloy has changed significantly after EMP exposure to the liquid metal. The microstructure of the untreated sample contains V dendrites, both in the form of the coarse inclusions, with sizes ranging from 50–100 μm, and smaller inclusions of about 50 μm. The main phase also exhibits a dendritic characteristic, in which dendritic cells have a size of 50–80 μm, and segregation of the second phase is observed in the inter-dendritic regions. These findings are consistent with previous reports [30,33]. The microstructure of the sample after EMP treatment is more homogeneous and does not contain a pronounced dendritic microstructure. Precipitation of primary and secondary phases in the treated $Al_{0.25}CoCrFeNiV$ sample has a lamellar "pearlite-like" microstructure. Instead of coarse primary V dendrites, cooperative lamellar precipitates of about 20–30 μm are observed.

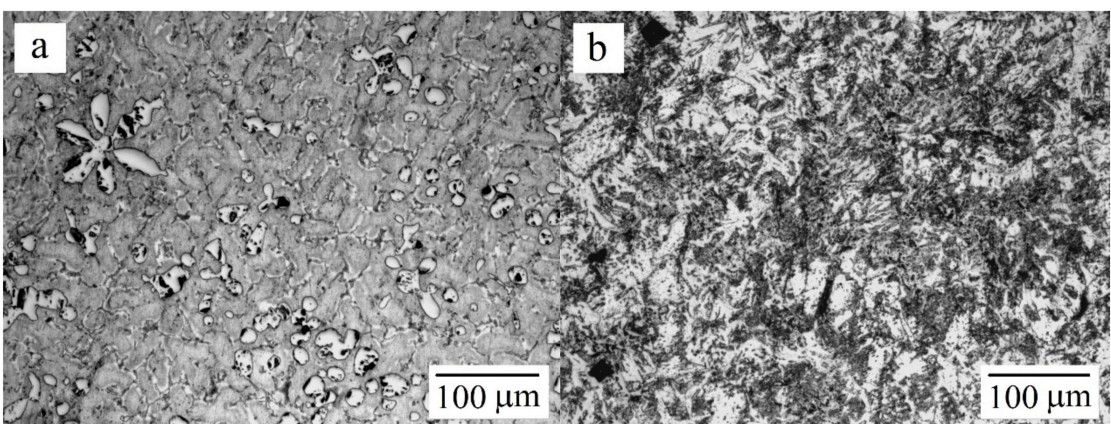

**Figure 2.** Optical microscopy microstructure of $Al_{0.25}CoCrFeNiV$ HEA; without EMP (**a**) and after EMP (**b**).

A similar EMP-effect on the size of cementite or graphite plates has already been reported for steels and cast irons [19–21]. Räbiger et al. [15] attributed such EMP-induced changes in the microstructure to the influence of the Lorentz force on the charged ions in the molten metal, which contributes to the strong convection in the molten metal. This, in turn, may result in unique microstructural patterns after crystallization. Moreover, the main cause of such microstructural changes is considered to be the emergence of mechanical forces in the melt due to the transformation of electromagnetic oscillations to ultrasonic oscillations during the solidification process, which causes the grinding of nucleation centers and mixing of the melt [20–22,31,32].

Ultrasonic waves may effectively modify the microstructure of the samples during melting and solidification. As reported by Arkin et al. [34], in order to create obvious positive effects on nucleation when ultrasonic waves flow through a metal melt, the electrodynamic pressure in the melt should be in the range of $1 \times 10^5$ to $4 \times 10^5$ Pa. The EMP-induced pressure in the melt can be calculated by the formula:

$$p = \frac{w \cdot (1 - R)}{v} \tag{1}$$

where $p$ is wave pressure ($N/m^2$), $w$ is specific power of impulse wave ($W/m^2$), $R$ is the reflection coefficient from the metal ($R = 0$ for total absorption, $R = 1$ for total reflection) and $v$ is wave speed (m/s). The wave speed in molten metals is in the order of $4 \times 10^3$ m/s [35].

With the pulsed excitation of oscillations, the pulsed power can be estimated according to the formula:

$$P = \frac{U^2}{r} \cdot (1 - K) \tag{2}$$

where $r$ is the cable wave impedance equal to 50 Ohm, $U$ is the generator voltage equal to 10 kV, and $K$ is the electromagnetic wave loss factor in lead wires and cables.

Considering that $K = 0.3$, the incident pulse power is obtained from formula (2), $P = 1.4 \times 10^6$ W. The area of the free surface of the metal in the used crucible with a diameter of 40 mm was equal to $1.3 \times 10^{-3}$ m$^2$. By substituting the obtained value of the pulse power related to the area and time into Equation (1), we estimated the pressure value in the melt in the order of $p = 2.7 \times 10^5$ Pa. This value is close in magnitude to the pressure values given by Arkin et al. [34] and Yu et al. [36] when the ultrasound is used to obtain a visible effect on the microstructure of metal samples. Thus, the performed calculation confirms the possible effect of ultrasonic vibrations on the HEA microstructure during irradiation of a metal melt with electromagnetic pulses.

SEM observations revealed some microstructural features of the samples (see Figure 3). As can be seen in Figure 3a–d, after the EMP, the particles of the third phase can be found in the grains of the solid solution, which precipitated in the form of needles with dimensions of about 10 µm in length and no more than 200 nm in width (Figure 3d). These particles were not observed in the untreated alloy (Figure 3b). Moreover, the morphology of V-rich dendrites changes from a flower-like to a rod-like structure upon exposure to the EMP. For a more detailed understanding of the distribution of elements over the phases, EDS mapping was carried out on the samples (Figure 4). The chemical composition of the phases was determined by an EDS analyzer (Table 1). It was found that V-rich dendrites do not contain Al, Fe, Co, and Ni elements. Instead, these elements are mainly present in solid solutions, which is consistent with their phase diagrams with vanadium [37–41]. At the same time, Cr exhibits an almost uniform distribution, consisting of a Cr–V phase diagram that is characterized by the unlimited solubility of two elements in liquid and solid states [42,43]. In addition, according to the indicated diagrams, the formation of a number of intermetallic compounds is possible in an Al–Co–Cr–Fe–Ni–V multicomponent system, and most likely, the EMP provoked the precipitation of one of these intermetallic compounds.

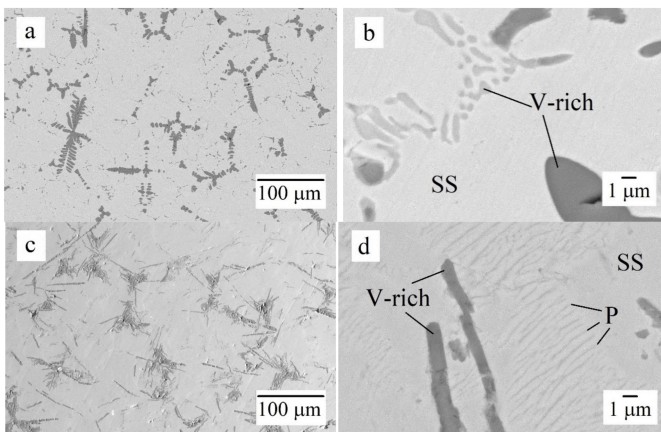

**Figure 3.** SEM microstructure of the Al$_{0.25}$CoCrFeNiV alloy without EMP (**a**,**b**), and after EMP (**c**,**d**).

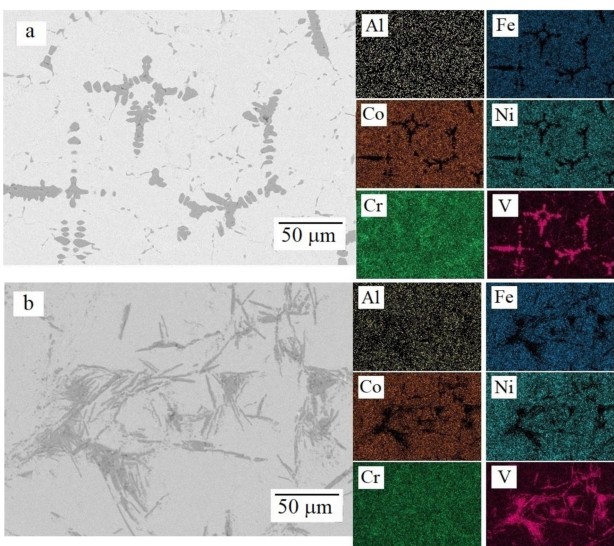

**Figure 4.** EDS mapping for $Al_{0.25}CoCrFeNiV$ alloy, (**a**) without the EMP and (**b**) after the EMP.

**Table 1.** EDS chemical composition of samples without and with the EMP (at.%). Av—average composition; SS—solid solution; V-rich—saturated V phase; P—precipitate in solid solution (see Figure 3).

| Alloy | | Al | Cr | Fe | Co | Ni | V | W |
|---|---|---|---|---|---|---|---|---|
| $Al_{0.25}CoCrFeNiV$ without EMP | Av | 4.69 | 19.21 | 18.88 | 19.01 | 18.94 | 19.16 | 0.11 |
| | SS | 4.45 | 18.96 | 21.15 | 21.90 | 21.93 | 11.10 | 0.51 |
| | V-rich | 0.87 | 8.91 | 0.95 | 1.09 | 0.88 | 87.30 | – |
| $Al_{0.25}CoCrFeNiV$ after EMP | Av | 4.47 | 19.60 | 19.02 | 19.36 | 18.42 | 19.00 | 0.13 |
| | SS | 4.87 | 17.54 | 20.11 | 22.14 | 21.52 | 13.21 | 0.61 |
| | V-rich | 0.31 | 16.19 | 2.21 | 2.37 | 1.91 | 77.01 | – |
| | P | 5.88 | 23.36 | 18.36 | 19.71 | 15.89 | 16.80 | – |

Taking into account the microstructure and mapping data (see Figures 3 and 4), as well as the numerical values from Table 1, we can note the formation of solid solution grains (SS) and the vanadium-enriched phase (V-rich) in $Al_{0.25}CoCrFeNiV$ HEA, as well as the presence of precipitated intermetallic compounds (P) in the alloy upon irradiation with the electromagnetic pulse.

According to Table 1, it can be stated that there was a slight dissolution of the W electrode into the melt, as its presence was detected in the composition of the solid solution; however, the content of W in the solid solution is ~0.5 at.%, and about 0.11–0.13 at.% in the average composition. Such an amount of the dissolved tungsten can be neglected when discussing the microstructure and properties of the samples.

### 3.2. X-ray Diffraction

The XRD spectra of the samples are shown in Figure 5. For the sample without EMP treatment, peaks related to the fcc solid solution and V-rich bcc phase are visible. After EMP treatment, in addition to these two phases, some peaks could be indexed as σ-phase with a tetragonal crystal lattice. The possibility of the formation of the σ-phase in a CoCrFeNiV alloy is indicated by Salishchev et al. [28]. The needle-like σ-phase has already been reported in $Al_{0.5}CoCrCuFeNiV$ HEA by Chen et al. [33].

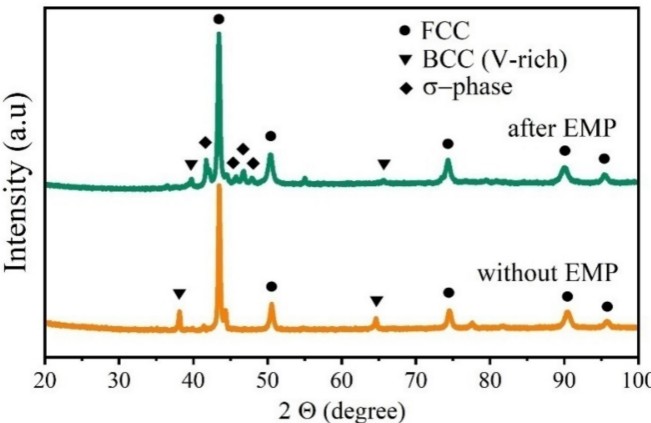

**Figure 5.** X-ray spectra of Al$_{0.25}$CoCrFeNiV HEA before and after EMP treatment.

According to our XRD analysis, the σ-phase is characterized by a tetragonal crystal lattice with parameters *a* = 8.816 Å and *c* = 4.571 Å, which coincides with the results reported by Salishchev et al. [28], where the σ-phase is characterized by a tetragonal lattice with parameters that can vary between *a* = 8.794–8.826 Å and *c* = 4.566–4.578 Å (depending on the degree of lattice distortion). Moreover, considering the interaction volume of electrons with the sample in EDS analysis (which is about 1 μm [44]), the precise chemical composition of the σ precipitates could not be determined.

The results of X-ray phase analysis are in agreement with the observed microstructure (Figures 3 and 4); therefore, it can be confirmed that the two-phase microstructure in the sample without EMP, changes to a three-phase microstructure after EMP exposure.

For HEAs, it is useful to calculate the entropy of mixing. According to Boltzmann's theory [45]:

$$\Delta S_{\mathrm{mix}} \ = \ -R \sum_{i\,=\,1}^{n} X_i \ln X_i \tag{3}$$

where *R* is the universal gas constant (*R* = 8.314 J/(mol·K)) and X is the atomic fraction of the elements in the HEA. Using data from Table 1, it is possible to calculate the entropy of mixing for each phase in Figure 5. The results of the calculation are given in Table 2. According to the obtained data, Al$_{0.25}$CoCrFeNiV HEA without EMP treatment is characterized by the presence of a high-entropy solid solution matrix, in which low-entropy V-rich dendrites are distributed. The EMP treatment made it possible to distribute the elements over the phases more evenly, which led to an increase in the entropy of mixing for the bcc (V-rich) phase. It should be noted that the σ-phase that is released after exposure to pulses can be also described as a high-entropy intermetallic phase. Recently, several high entropy intermetallics with different crystal structures have been successfully fabricated [46].

**Table 2.** The calculated entropy of mixing for each phase in the investigated Al$_{0.25}$CoCrFeNiV HEA (see Figure 5).

| Phase | Without EMP | | After EMP | | |
|---|---|---|---|---|---|
| | SS (FCC) | V-Rich (BCC) | SS (FCC) | V-Rich (BCC) | P (σ-Phase) |
| $\Delta S_{\mathrm{mix}}$, J · mol$^{-1}$ · K$^{-1}$ | 14.29 | 4.24 | 14.45 | 6.34 | 14.38 |

Therefore, considering the $\Delta S_{\mathrm{mix}}$ microstructure and phase composition, Al$_{0.25}$CoCrFeNiV can be described as a composite material in which a low-entropy strengthening phase is distributed in a high-entropy matrix phase. The EMP treatment creates an even more unique composite, because an additional high-entropy intermetallic phase precipitates in the high-entropy matrix, owing to the high-power electromagnetic pulses.

The effect of EMP on the microstructural evolution of the alloy can be explained by the amount of the energy introduced into the melt. The energy supplied to the melt can be estimated as follows. In each pulse, 4.5 MW power is transferred into the melt from the generator. Part of this power is lost due to the resistance of the lead wires and cable. The loss factor measured in the cold state for this study was $K = 0.3$. Thus, taking into account the losses, the power supplied to the melt in one pulse is about $P = 3.15$ MW. If we take into account the pulse frequency of 1000 Hz, and the processing time of the melt $\tau = 300$ s (5 min), then the energy $E$ transferred into the melt can be determined by the formula:

$$E = P \cdot \tau \tag{4}$$

which was equal to $9.45 \times 10^{11}$ J. The specific energy in this case turned out to be equal to $2.36 \times 10^{13}$ J/kg. Obviously, this energy is sufficient to initiate phase transformations.

### 3.3. Microhardness and Gas-Abrasive Wear Resistance

The microhardness of the samples was $343 \pm 10$ HV$_{0.3}$ without EMP, and increased to $553 \pm 15$ HV$_{0.3}$ after EMP. A microhardness index of $176 \pm 5$ HV$_3$ and 151 HV has been reported for as cast Al$_{0.3}$CoCrFeNi [47] and Al$_{0.25}$CoCrFeNi [48] HEAs, respectively. Moreover, it has been shown that the microhardness of Al$_{0.25}$CoCrFeNi can be improved to 260 HV after annealing and rolling [48]. It can be noted that adding vanadium to the Al$_x$CoCrFeNi base alloy significantly increases its microhardness value. Moreover, after EMP exposure on the melt and the corresponding changes in the microstructure of Al$_{0.25}$CoCrFeNiV alloy, the microhardness of the sample increased by 61%. Such microhardness values are comparable to that reported for the CoCrFeNiV alloy (524 HV$_{0.25}$), which is attributed to the formation of the σ–phase [28]. Chen et al. [33] also noted a sharp increase of up to 650 HV in the microhardness of the Al$_{0.5}$CoCrCuFeNiV HEA, due to the appearance of the σ-phase needles.

Figure 6 shows the mass loss of Al$_{0.25}$CoCrFeNiV HEA samples as a function of the exposure time of the samples' surface to the flow of abrasive particles. As shown in the graphs, the sample after the EMP exhibits a higher resistance against gas-abrasive wear. After 20 min of exposure to a mixture of air with an abrasive particle under pressure, the maximum weight loss for the sample without the EMP was 1.00 g/cm$^2$, whereas it was about 0.76 g/cm$^2$ for the sample after the EMP. The curve for the sample without EMP is almost linear, which indicates a uniform structure over the entire volume of the alloy. For the sample after EMP, there are deviations from a linear dependence, which can be associated with the formation of the hard σ-phase with a non-uniform distribution. Moreover, the brittleness of the σ-phase may affect the resistance of the sample to gas-abrasive wear [28,33].

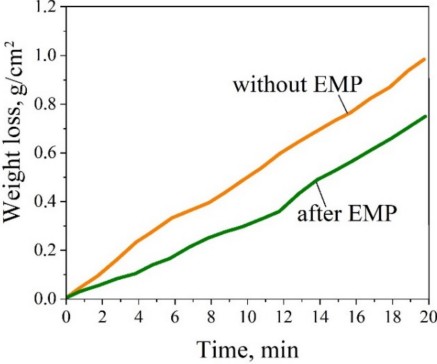

**Figure 6.** Weight loss of Al$_{0.25}$CoCrFeNiV HEA as a function of exposure time to the flow of abrasive particles.

It is widely accepted that developing a composite material with more reinforcement particles and with a more uniform distribution can improve the mechanical properties of

the samples. Here, XRD and microstructural observation clearly indicated that the EMP improves the distribution of the reinforcement particles and also triggers the formation of a sigma phase, which obviously has beneficial effects on the microhardness and wear resistance of the samples; however, according to Figure 6, the difference between the mass loss of the samples is much smaller (in percentage) than the difference between microhardness values. This may suggest the possible embrittlement of the samples upon precipitation of the hard σ-phase. Considering the needle-like shape of the σ-phase, it can be assumed that abrasive particles break needles (stress concentrator centers in the alloy), and over time, the solid solution matrix also starts being actively chipped. An upward jump in the microhardness, and a moderate improvement in the wear resistance of the samples containing σ-phase, have been also reported by Chen et al. [33].

Figure 7 shows the surface morphology of the samples after 20 min of exposure to abrasive particles. It can be seen that the surface of the sample without the EMP was subjected to somewhat more intense wear than that of the sample with the EMP. The surface of the EMP treated sample is more even, without obvious "holes", and there are quite extensive fields without visible abrasive action. The size of the lump and pits could be estimated at around 10–20 μm, which is two times less than that of the sample without EMP treatment (for which the pit size is about 30–50 μm). Moreover, there are wider and deeper wear traces on the surface of the sample without EMP. The gas-abrasive behavior of the sample treated with EMP is attributed to its more uniform microstructure and the presence of hard σ-phase needles in its microstructure [28,33].

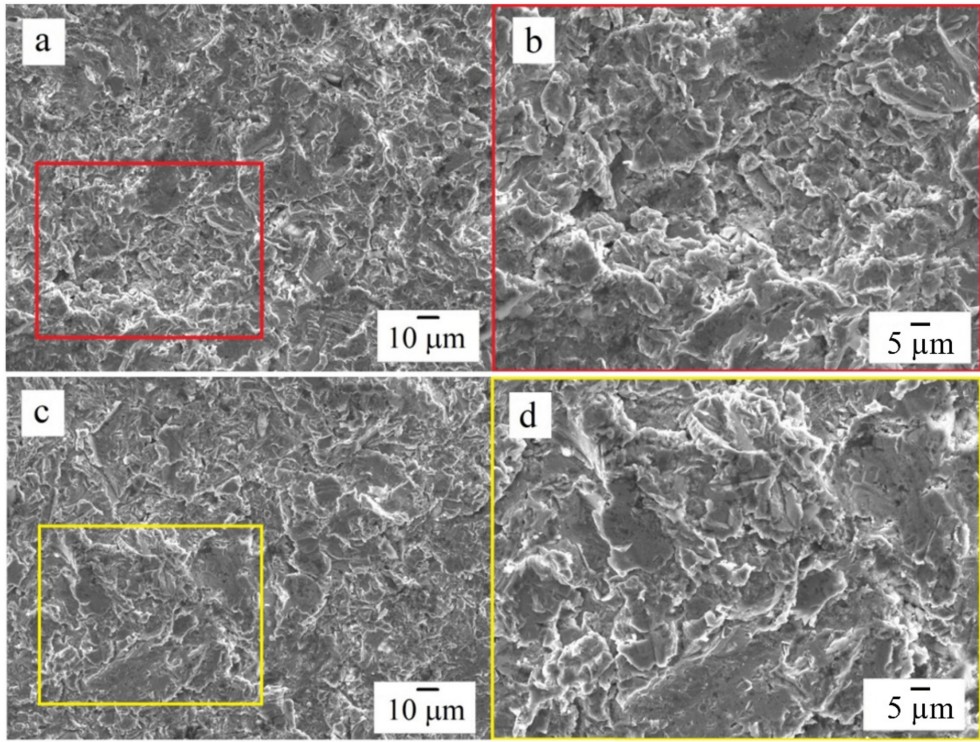

**Figure 7.** Surface morphology of $Al_{0.25}CoCrFeNiV$ HEA after 20 min of gas-abrasive wear for the samples without (**a**,**b**) and with (**c**,**d**) EMP.

## 4. Conclusions

The effect of EMP on the microstructure, microhardness, and gas-abrasive wear behavior of $Al_{0.25}CoCrFeNiV$ HEA was studied. The untreated sample is characterized by coarse primary vanadium dendrites with a size in the range of 50–100 μm distributed in the solid solution matrix. The sample after the EMP exhibits a lamellar "pearlite-like" microstructure with rod-like vanadium-rich precipitates, which are 20 to 30 μm in length and about 5 μm thick. Moreover, in the sample after the EMP, the formation of a needle-like σ-phase

with a tetragonal crystal lattice was revealed. The microhardness of the $Al_{0.25}CoCrFeNiV$ increased from $343 \pm 10$ $HV_{0.3}$ (for the sample without EMP) to $553 \pm 15$ $HV_{0.3}$ (for the sample after the EMP), which is ascribed to the microstructure refinement and σ-phase precipitation. The sample after the EMP also exhibits higher resistance to gas-abrasive wear. Finally, the formation and content of the σ-phase could likely be adjusted by refining the EMP treatment, which is an assumption that needs to be examined in future studies.

**Author Contributions:** Conceptualization, O.S. and N.S.; methodology, N.S. and E.T.; validation, O.S., N.S., A.O.M. and E.T.; formal analysis, V.K.; investigation, N.S. and O.S.; resources, A.O.M.; data curation, V.M.; writing—original draft preparation, O.S.; writing—review and editing, A.O.M. and O.S.; visualization, N.S.; supervision, V.K.; project administration, V.K. and N.S. All authors have read and agreed to the published version of the manuscript.

**Funding:** This research was funded by the Ministry of Science and Higher Education of the Russian Federation as part of the project FENU-2020-0020.

**Institutional Review Board Statement:** Not applicable.

**Informed Consent Statement:** Not applicable.

**Data Availability Statement:** The raw/processed data required to reproduce these findings are available from the corresponding author upon reasonable request.

**Conflicts of Interest:** The authors declare no conflict of interest.

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
