# Peer review of "Effect of Electromagnetic Pulses on the Microstructure and Abrasive Gas Wear Resistance of Al0.25CoCrFeNiV High Entropy Alloy"

_coatings, doi:10.3390/coatings12050688_

Round 1

Reviewer 1 Report

  1. Authors stated that “The loss factor measured in the cold state for this study was K = 0.3”. How was K value measured and was it suitable in the melting state?
  2. The wear traces should be compared quantificationally by scanning the profiles.
  3. The direct evidenceof the effect of σ-phase on abrasive resistance is still lacking. The detailed abrasive-resistant mechanism need to be further explored.

Author Response

Dear Editor,

Thank you for giving us the opportunity to revise our review manuscript and elaborate its structure and quality. We sincerely thanks the reviewer comments and his/her suggestions to improve the quality and readability of this manuscript. We enjoyed considering the comments and elaborating our manuscript quality. Moreover, we tried to correct all the typo and grammatical mistakes all through the manuscript.

We highlighted all the changes through the manuscript and presented a list of detailed corrections and response to the comments.

We look forward to your positive response.

Sincerely,

Ahmad Ostovari Moghaddam

Reviewer 1 Comments and Suggestions for Authors

  1. Authors stated that “The loss factor measured in the cold state for this study was K = 0.3”. How was K value measured and was it suitable in the melting state?

Thank you for your comments, which helped make our manuscript much better.

The loss factor K was measured by using a picosecond pulsed reflectometer P5-15. The operating frequency band of the instrument is 0–5 GHz. The essence of the measure-ment is to determine the reflection from the end of the line that includes the cable (supplies energy from the generator), wires to the electrode, electrode and wires to the crucible. The measured value in the cold state (without immersing the electrodes into the melt) was 0.3. This means that 0.3 of the power supplied from the generator is lost in the line elements (cable, wires, electrode). Consequently, the remaining 70% of the power gets into the melt.

The amount of losses depends only on the linear dimensions, the material of the specific elements of the line and does not depend on the operating conditions. Those 70% of the remaining pulse power is transfer into the melt, even into any other medium to which the electrodes are connected.

Corresponding changes have been made to the text of the manuscript.

2. The wear traces should be compared quantificationally by scanning the profiles.

As the reviewer suggested, profile scanning is commonly used for the analysis of the materials subjected to abrasion wear. While a quantification analysis of wear traces by scanning the profiles could add more interesting data to the manuscript, we do not have access to such a scanning profilometer to conduct this test. We apologize the reviewer for this. However, the technique employed in this work by using an abrasive blasting booth assumes the impact of the abrasive on a sufficiently large area and covers the entire area of the tested sample surface. For such a technique, profile scanning can become a lengthy process with a high probability of error. Usually, according to the standard methods used (GOSTs 23.207-79 and 23.208-79), in this case it is more correct to use the value of weight loss per unit area of the test sample. Fig. 7 is shown for clarity of the surface obtained after gas-abrasive wear.

3. The direct evidence of the effect of σ-phase on abrasive resistance is still lacking. The detailed abrasive-resistant mechanism need to be further explored.

The reviewer is absolutely right. Unfortunately, due to the worn-rugged surface of the samples, we couldn’t find the trace of sigma phase on the worn surface of the samples. At this stage we have revealed the changes in the microstructural and phase composition of Al0.25CoCrFeNiV alloy under the action of electromagnetic pulses. A change in the morphology of vanadium precipitates and the formation of a strengthening σ-phase should lead to a significant increase in the resistance of the sample surface to abrasive action. We tried to add more discussion and provided suitable references to cover this aspect of the manuscript. These sentences were added to the manuscript:

“It is widely accepted that developing a composite material with more reinforcement particles and with more uniform distribution can improve mechanical properties of the samples. Here, XRD and microstructural observation clearly indicated that EMP improves the distribution of the reinforcement particles and also triggers the formation of a sigma phase, which obviously has beneficial effects on the microhardness and wear resistance of the samples. However, according to Fig. 6, the difference in mass losses of the samples is much smaller (in percentage) than the difference in micro-hardness values. This may suggest the possible embrittlement of the samples upon precipitation of the hard σ-phase. Considering the needle-like shape of the σ-phase, it can be assumed that abrasive particles break needles (stress concentrator centers in the alloy), and over time, the solid solution matrix also begins to be actively chipped. An upward jump in the microhardness and a moderate improvement in the wear resistance of the samples containing σ-phase have been also reported by Chen et al. [33].”

All changes are highlighted in the text of the manuscript with a yellow marker.

Reviewer 2 Report

This paper aims to study on the effect of electromagnetic pulses on the microstructure and abrasive gas wear resistance of Al0.25CoCrFeNiV high entropy alloy.

I believe that this piece of research does not meet the quality standards of “Journal of Coatings”, and therefore do not recommend its publication in this Journal. This is due to the lack of novelty and poor discussion of the present manuscript.

The effect of electromagnetic pulses on the microstructure has been extensively studied by other researchers,  and the authors have used some of these available references in their study. Added to this, how did the authors conclude that the superior gas-abrasive behavior of the sample treated with EMP is related to its more uniform microstructure and the presence of hard σ-phase needles in its microstructure? The mechanism must be explained in detail.   

Author Response

Dear Editor,

Thank you for giving us the opportunity to revise our review manuscript and elaborate its structure and quality. We sincerely thanks the reviewer comments and his/her suggestions to improve the quality and readability of this manuscript. We enjoyed considering the comments and elaborating our manuscript quality. Moreover, we tried to correct all the typo and grammatical mistakes all through the manuscript.

We highlighted all the changes through the manuscript and presented a list of detailed corrections and response to the comments.

We look forward to your positive response.

Sincerely,

Ahmad Ostovari Moghaddam

Reviewer 2

This paper aims to study on the effect of electromagnetic pulses on the microstructure and abrasive gas wear resistance of Al0.25CoCrFeNiV high entropy alloy.

I believe that this piece of research does not meet the quality standards of “Journal of Coatings”, and therefore do not recommend its publication in this Journal. This is due to the lack of novelty and poor discussion of the present manuscript.

The effect of electromagnetic pulses on the microstructure has been extensively studied by other researchers, and the authors have used some of these available references in their study. Added to this, how did the authors conclude that the superior gas-abrasive behavior of the sample treated with EMP is related to its more uniform microstructure and the presence of hard σ-phase needles in its microstructure? The mechanism must be explained in detail.

Answer for Reviewer 2:

Thank you for your comments, which helped make our manuscript much better.

The available works on pulsed processing of metal melts (as shown in our review Shaburova N., Krymsky V., Ostovari Moghaddam A. Theory and Practice of Using Pulsed Electromagnetic Processing of Metal Melts. Materials, 2022, Vol. 15, pp. 1235. DOI: 10.3390/ma15031235) describe the use of various methods for generating electromagnetic pulses. All of them, as a rule, are similar in basic characteristics, duration, frequency, power. A feature of the EMP pulse action in our work is the unique combination of short pulse duration (1 ns), pulse repetition rate of 1 kHz, and calculated pulse power of 4.5 MW. Such characteristics are provided by using a special type of generator, which are assembled by hand at two enterprises in the world.

The main part of the literature works is devoted to the study of the effect of EMP on non-ferrous (copper, aluminum, magnesium, etc.) alloys; indeed, a whole array of data has been accumulated there.

But for HEAs, the study of the effect of EMP on the microstructure has not yet been sufficiently studied (there are only about 4-5 works). Moreover, for Al0.25CoCrFeNiV composition, such a study was carried out by us for the first time. So, from our point of view, this study is novel and of interest to experts in materials science and can be used to create materials with a unique microstructure and improved properties.

It is widely accepted that developing a composite material with more reinforcement particles and with more uniform distribution can improve mechanical properties of the samples. Here, XRD and microstructural observation clearly indicated that EMP improves the distribution of the reinforcement particles and also trigger the formation of a sigma phase, which obviously has beneficial effects on the microhardness and wear resistance of the samples. We added more discussions about the effect of microstructure on the gas-abrasive behavior of the sample. Same arguments were also added to the manuscript:

“It is widely accepted that developing a composite material with more reinforcement particles and with more uniform distribution can improve mechanical properties of the samples. Here, XRD and microstructural observation clearly indicated that EMP improves the distribution of the reinforcement particles and also triggers the formation of a sigma phase, which obviously has beneficial effects on the microhardness and wear resistance of the samples. However, according to Fig. 6, the difference in mass losses of the samples is much smaller (in percentage) than the difference in microhardness values. This may suggest the possible embrittlement of the samples upon precipitation of the hard σ-phase. Considering the needle-like shape of the σ-phase, it can be assumed that abrasive particles break needles (stress concentrator centers in the alloy), and over time, the solid solution matrix also begins to be actively chipped. An upward jump in the microhardness and a moderate improvement in the wear resistance of the samples containing σ-phase have been also reported by Chen et al. [33].”

Corresponding changes have been highlighted in green.

Reviewer 3 Report

Based on the comprehensive analysis of the relevant literature, the influence of electromagnetic pulses on the microstructure and abrasive gas wear resistance of Al0.25CoCrFeNiV high entropy alloy is systematically investigated. The paper has the outstanding innovation and reasonable analysis. However, there are still some problems for authors to deal with before publication.

  1. Please improve the abstract to cover the work's importance and main ideas.
  2. Please improve the quality of images using the high resolution images. It is somehow difficult to read the information in some figures, for example, Figure 2b.
  3. The sentence "For high entropy alloys" in the line 226 is incorrectly expressed, because the abbreviation for high entropy alloys already exists in the previous text. It should be changed to “For HEAs”.
  4. According to the author's description, Figure 7(b, d) should be the enlarged area of Figure 7(a, c), but the scales of the four figures are the same, please correct.
  5. From line 291 to 293, the authors argue that “the excellent gas-abrasive behavior of the sample treated with EMP is attributed to its more uniform microstructure and the presence of hard σ-phase needles in its microstructure”. However, there is no any direct evidence for the full text to prove the view. It is recommended that the author add corresponding references.
  6. The point of the paper is that the microstructure and the σ-phase in the Al0.25CoCrFeNiV HEA after EMP improve its gas-abrasive wear resistance, while the section 3.1 mainly characterizes the microstructure changes and the V-rich precipitates of the HEA before and after EMP. There are few descriptions about the σ-phase, and its microstructure, element composition and lattice parameters are unknown. The basic information of the σ-phase should be added.
  7. The conclusions are too long and cumbersome, and therefore they should be simplified.

Author Response

Dear Editor,

Thank you for giving us the opportunity to revise our review manuscript and elaborate its structure and quality. We sincerely thanks the reviewer comments and his/her suggestions to improve the quality and readability of this manuscript. We enjoyed considering the comments and elaborating our manuscript quality. Moreover, we tried to correct all the typo and grammatical mistakes all through the manuscript.

We highlighted all the changes through the manuscript and presented a list of detailed corrections and response to the comments.

We look forward to your positive response.

Sincerely,

Ahmad Ostovari Moghaddam

Reviewer 3

Based on the comprehensive analysis of the relevant literature, the influence of electromagnetic pulses on the microstructure and abrasive gas wear resistance of Al0.25CoCrFeNiV high entropy alloy is systematically investigated. The paper has the outstanding innovation and reasonable analysis. However, there are still some problems for authors to deal with before publication.

Thank you for your comments, which helped make our manuscript much better. All changes are highlighted in the text of the manuscript with a turquoise marker.

1. Please improve the abstract to cover the work's importance and main ideas.

This sentence was added to the abstract to cover the work's importance and main ideas: “Finally, EMP is introduced as an effective route to modify the microstructure and phase formation of cast HEAs, which in turn opens up broad horizons for fabricating cast samples with tailorable microstructure and improved properties.”

2. Please improve the quality of images using the high resolution images. It is somehow difficult to read the information in some figures, for example, Figure 2b.

We tried to improve the quality of images further. All images are obtained on high-tech equipment and have the highest possible resolution. It should be clarified that the SEM images of the microstructure (Figs. 3, 4) were obtained at high magnifications ×2000-3000. On Fig. 2b microstructure obtained on an optical microscope at a magnification ×200. Since lamellar perlite-like precipitates in the EMP-treated sample are very dispersed (they have a small interlamellar distance, as shown in Fig. 3d), at low magnifications they are not resolved as separate elements, but look like a single conglomerate (by analogy with pearlite in steels).

3. The sentence "For high entropy alloys" in the line 226 is incorrectly expressed, because the abbreviation for high entropy alloys already exists in the previous text. It should be changed to “For HEAs”.

The HEAs abbreviation is now used.

4. According to the author's description, Figure 7(b, d) should be the enlarged area of Figure 7(a, c), but the scales of the four figures are the same, please correct.

We apologize for this mistake. Corresponding changes have been made to the figures.

5. From line 291 to 293, the authors argue that “the excellent gas-abrasive behavior of the sample treated with EMP is attributed to its more uniform microstructure and the presence of hard σ-phase needles in its microstructure”. However, there is no any direct evidence for the full text to prove the view. It is recommended that the author add corresponding references.

Thank you for your suggestion. We added related references to support our claim.

6. The point of the paper is that the microstructure and the σ-phase in the Al0.25CoCrFeNiV HEA after EMP improve its gas-abrasive wear resistance, while the section 3.1 mainly characterizes the microstructure changes and the V-rich precipitates of the HEA before and after EMP. There are few descriptions about the σ-phase, and its microstructure, element composition and lattice parameters are unknown. The basic information of the σ-phase should be added.

These sentences were added to the manuscript:

“Considering that there is no phase diagram for such a multicomponent alloy in the literature, and based on the data from Table 1 and literature [28, 33], it can be assumed that the precipitated σ-phase has a composition of (Al1/4CoFeNi)(Cr3/2V). According to Salishchev et al. [28], the σ-phase in Co-Cr-Fe-Ni-Mn-V HEAs is characterized by a tetragonal lattice with parameters that can vary between a = 8.794–8.826 Ǻ and c = 4.566–4.578 Ǻ (depending on the degree of lattice distortion).”

7. The conclusions are too long and cumbersome, and therefore they should be simplified.

The Conclusions were rewritten as:

“The effect of EMP on the microstructure, microhardness and gas-abrasive wear behavior of Al0.25CoCrFeNiV HEA was studied. The untreated sample is characterized by coarse primary vanadium dendrites with a size in the range of 50–100 µm distributed in the solid solution matrix. The sample after EMP exhibits a lamellar "pearlite-like" microstructure with rod-like vanadium-rich precipitates with 20 to 30 μm length and about 5 μm thick. Moreover, in the sample after EMP the formation of a needle-like σ-phase with a tetragonal crystal lattice was revealed. The microhardness of the Al0.25CoCrFeNiV increased from 343 ± 10 HV0.3 (for the sample without EMP) to 553 ± 15 HV0.3 (for the sample after EMP), which is ascribed to the microstructure refinement and σ-phase precipitation. The sample after EMP exhibits also higher resistance to gas-abrasive wear. Finally, the formation and content of σ-phase could be likely adjusted by tuning the EMP treatment, an assumption that need to be examined in future studies.

Round 2

Reviewer 1 Report

The manuscript has been carefully revised. It can be accepted.

Reviewer 2 Report

Unfortunately, the severe lack of novelty of the manuscript is not improved. The whole story, the deduction, and the discussion have serious flaws, and did not improve.  All the main sentences and results are cited from other references. As I already mentioned, the sigma phase and its role is the main result of the manuscript. However, there is not enough evidence such as observation, ... to show that.

Also, the organization of the manuscript is very poor. Therefore, this manuscript does not deserve publication.